# $z$-SignFedAvg: A Unified Sign-based Stochastic Compression for Federated Learning

## Abstract

Federated learning is a promising privacy-preserving distributed learning paradigm
but suffers from high communication cost when training large-scale machine
learning models. Sign-based methods, such as SignSGD [Bernstein et al., 2018],
have been proposed as a biased gradient compression technique for reducing
the communication cost. However, sign-based compression could diverge under
heterogeneous data, which motivate developments of advanced techniques, such
as the error-feedback method and stochastic sign-based compression, to fix this
issue. Nevertheless, these methods still suffer significantly slower convergence
rate than uncompressed algorithms. Besides, none of them allow local multiple
SGD updates like FedAvg [McMahan et al., 2017]. In this paper, we propose a
novel noisy perturbation scheme with a general symmetric noise distribution for
sign-based compression, which not only allows one to flexibly control the tradeoff
between gradient bias and convergence performance, but also provides a unified
viewpoint to existing sign-based methods. More importantly, we propose the very
first sign-based FedAvg algorithm ($z$-SignFedAvg). Theoretically, we show that
$z$-SignFedAvg achieves a faster convergence rate than existing sign-based methods
and, under the uniformly distribtued noise, can even enjoy the same convergence
rate as its uncompressed counterpart. Extensive experiments are conducted to
demonstrate that our proposed $z$-SignFedAvg can achieve competitive empirical
performance on real datasets.

## 1   Introduction

In this paper, we consider the Federated Learning (FL) network with one parameter server and $n$
clients [McMahan et al., 2017, Li et al., 2020a], with the focus on solving the following distributed
learning problem

$$\min_{x \in \mathbb{R}^d} f(x) = \frac{1}{n} \sum_{i=1}^n f_i(x), \tag{1}$$

where $f_i(\cdot)$ is the local objective function for the $i$-th client, for $i = 1, \ldots, n$. Throughout this
work, we assume that each $f_i$ is smooth and possibly non-convex. The local objective functions are
generated from the local dataset owned by each client. When designing distributed algorithms to solve
(1), a crucial aspect is the communication efficiency since each client needs to transmit their local
gradients to the server frequently [Li et al., 2020a]. As one of the most popular FL algorithms, the
federated averaging (FedAvg) algorithm [McMahan et al., 2017, Konečnỳ et al., 2016] considers local
multiple SGD updates with periodic communications to reduce the communication cost. Another
way is to compress the local gradients before sending them to the server [Li et al., 2020a, Alistarh
et al., 2017, Reisizadeh et al., 2020]. Among the existing compression methods, a simple yet elegant
technqiue is to take the sign of each coordinate of the local gradients, which requires only one bit for

Submitted to 36th Conference on Neural Information Processing Systems (NeurIPS 2022). Do not distribute.

transmitting each coordinate. For any $x \in \mathbb{R}$, we define the sign operator as: $\text{Sign}(x) = 1$ if $x \geq 0$ and $-1$ otherwise.

Recently, optimization algorithms with sign-based compression have attracted much attention as they enjoy a great communication efficiency while still achieving comparable empirical performance as uncompressed algorithms [Bernstein et al., 2018, Karimireddy et al., 2019, Safaryan and Richtárik, 2021]. However, for distributed learning, especially the scenarios with heterogeneous data, i.e., $f_i \neq f_j$ for every $i \neq j$, a naive application of the sign-based algorithm cannot guarantee convergence [Karimireddy et al., 2019, Chen et al., 2020a, Safaryan and Richtárik, 2021]. There are mainly two approaches to fix this issue in the existing literature. The first one is the stochastic sign-based method, which intoduces stochasticity into the sign operation [Jin et al., 2020, Safaryan and Richtárik, 2021, Chen et al., 2020a], and the second one is the Error-Feedback (EF) method [Karimireddy et al., 2019, Vogels et al., 2019, Tang et al., 2019]. However, these works are still unstastifactory. Specifically, first, the theoretical convergence rates of these algorithms are still worse than uncompressed algorithms like [Ghadimi and Lan, 2013, Yu et al., 2019]. Second, none of them allows the clients to have multiple local SGD updates within one communication round like the FedAvg. This work aims at addresing these issues and closing the gaps for sign-based methods. A detailed review for related works is given in Appendix A.

**Main contributions.** Our contributions are summarized as follows.

(1) **A unified family of stochastic sign operators.** We show an intriguing fact: The bias brought by the sign-based compression can be flexibly controlled by injecting a proper amount of random noise before the sign operation. In particular, our analysis is based on a novel noisy perturbation scheme with a general symmetric noise distribution, and therefore provides a unified viewpoint and incorporates existing stochastic sign-based methods, including [Jin et al., 2020, Safaryan and Richtárik, 2021, Chen et al., 2020a], as special instances.

(2) **The first sign-based federated averaging algorithm.** In contrast to the existing sign-based shcemes which do not allow local multiple SGD updates within one communication round, based on the proposed stochastic sign-based compression, we design a novel family of sign-based federated averaging algorithms ($z$-SignFedAvg) that can achieve the best of both worlds: communication efficiency and convergence performance.

(3) **New theoretical convergence rate analyses.** By a clever use of the asymptotic unbiasedness property of the stochastic sign-based compression, we derive a series of theoretical results which exhibit better convergence rate than the existing sign-based methods. Moreover, we show that by injecting a suffciently large uniform noise, our proposed algorithm can have a matching convergence rate with the uncompressed algorithms.

**Organization of this paper.** In Section 2, the proposed general noisy perturbation scheme for the sign-based compression and its key propoerty about asymptotic unbiasedness are presented. Inspired by this result, the main algorithms are developed in Section 3 together with their convergence analyses under different noise distribution parameters. We evaluate our proposed algorithms on real datasets and benchmark with existing FL methods in Section 4. Finally, conclusions are drawn in Section 5.

**Notations.** For any $x \in \mathbb{R}^d$, we denote $x(j)$ as the $j$-th element of the vector $x$. We define the $\ell_p$ norm for any $p \geq 1$ as $\|x\|_p = (\sum_{j=1}^{d} |x(j)|^p)^{\frac{1}{p}}$. We denote that $\|\cdot\| = \|\cdot\|_2$, and $\|x\|_\infty = \max_{j \in \{1, \dots, d\}} |x(j)|$. For any function $f(x)$, we denote $f^{(k)}(x)$ as its $k$-th derivative, and for a vector $x = [x(1), \dots, x(d)]^\top \in \mathbb{R}^d$, we define $\text{Sign}(x) = [\text{Sign}(x(1)), \dots, \text{Sign}(x(d))]^\top$.

## 2 Sign operator with symmetric and zero-mean noise is asymptotically unbiased.

In this section, we introduce a general noisy perturbation scheme for the sign-based compression and analyze its asymptotic unbiasedness. The result serves as the foundation of the proposed algorithm designs in subsequent sections. Specifically, let us consider the following family of noise distribution parameterized by a postive interger $z \in \mathbb{Z}_+$.

**Definition 1** ($z$-distribution). *A random variable $\xi_z$ follows $z$-distribution if its p.d.f is*

$$p_z(t) = \frac{1}{2\eta_z} e^{-\frac{t^{2z}}{2}}, \tag{2}$$

85    *where $\eta_z = 2^{\frac{1}{2z}}\Gamma\left(1 + \frac{1}{2z}\right)$ and $\Gamma(z) = \int_0^{+\infty} t^{z-1}e^{-t}dt$ is the Gamma function.*

86    It can be verified that (2) is a valid p.d.f. When $z = 1$, it corresponds to the standard Gaussian
87    distribution. In addition, one can also show that (2) converges to the uniform distribution when
88    $z \to +\infty$ (Lemma 2 in Appendices). This family of distribution has a nice property that can be
89    leveraged to bound the bias caused by the sign compression, as stated in the following lemma.

90    **Lemma 1.** *For any $x \in \mathbb{R}^d$ and $\sigma > 0$,*

$$\left\|\eta_z\sigma\mathbb{E}\left[\mathrm{Sign}(x + \sigma\xi_z)\right] - x\right\|^2 \leq \frac{\|x\|_{4z+2}^{4z+2}}{4(2z+1)^2\sigma^{4z}}, \tag{3}$$

91    *where $\xi_z(1), ..., \xi_z(d)$ follow the i.i.d. z-distribution.*

92    **Remark 1.** *One can see that the RHS of (3) involves the term $\left(\|x\|_{4z+2}/\sigma\right)^{4z}$. Therefore, as long as*
93    *$\sigma > \|x\|_\infty$, the LHS of (3) converges to zero when $z \to +\infty$. Since Lemma 2 implies that $\xi_\infty$ is a*
94    *vector whose elements follow i.i.d uniform distribution at $[-1, 1]$, we obtain $\sigma\mathbb{E}\left[\mathrm{Sign}(x + \sigma\xi_\infty)\right] = x$*
95    *as long as $\sigma > \|x\|_\infty$. It is interesting to remark that the stochastic sign operators proposed in [Jin*
96    *et al., 2020, Safaryan and Richtárik, 2021] is exactly the sign operator injected with a sufficient*
97    *amount of uniform noise.*

## 3   Stochastic sign-based federated averaging with $z$-distribution.

99    In this section, based on the anaysis in Section 2, we propose the following sign-based federated
100   averaging algorithm, termed as $z$-SignFedAvg. The algorithm details are stated in Algortihm 1. Note
101   that in practice, we only implement $z$-SignFedAvg with $z = 1$ and $z = +\infty$ which correspond to
102   the Gaussian distribution and uniform distribution, respectively. This is because, to the best of our
103   knowledge, there is currently no efficient way to sampling from the distribution $p_z(t)$ with other
104   choices of $z$. Nevertheless, we are interested in the convergence properteis of $z$-SignFedAvg for a
105   general positive interger $z$ since it provides better insights on how $z$ impacts the convergence rate.

---

**Algorithm 1** $z$-SignFedAvg (or $z$-SignSGD when $E = 1$)

---

**Require:** Total communication rounds $T$, Number of local steps $E$, Number of clients $n$, Clients
     stepsize $\gamma$, Server stepsize $\eta$, Noise coefficient $\sigma$, parameter of noise distribution $z$.
 1: Initialize $x_0$ and for $i = 1, ..., n$.
 2: **for** $t = 1$ to $T$ **do**
 3:    **On Clients:**
 4:    **for** $i = 1$ to $n$ in parallel **do**
 5:      $x_{t-1,0}^i = x_{t-1}$
 6:      **for** $s = 1$ to $E$ **do**
 7:        $g_{t-1,s}^i = g_i(x_{t-1,s-1}^i)$, where $g_i(\cdot)$ is the minibatch gradient oracle of the $i$-th client.
 8:        $x_{t-1,s}^i = x_{t-1,s-1}^i - \gamma g_{t-1,s}^i$.
 9:      **end for**
10:      $\Delta_{t-1}^i = \mathrm{Sign}\left(\frac{x_{t-1} - x_{t-1,E}^i}{\gamma} + \sigma\xi_z\right)$, where $\xi_z(1), ..., \xi_z(d) \sim p_z(t)$ i.i.d.
11:      Send $\Delta_{t-1}^i$ to the server.
12:    **end for**
13:    **On Server:**
14:      $x_t = x_{t-1} - \eta\gamma\frac{1}{n}\sum_{i=1}^n \Delta_{t-1}^i$.
15:    Send $x_t$ to the clients.
16: **end for**
17: **return** $x_T$.

---

106   We first state the following standard assumptions.

107    **Assumption 1.** *We assume that each $f_i(x)$ has the following properties:*

108      *A.1 The minibatch gradient is unbiased and has bounded variance, i.e., $\mathbb{E}[g_i(x)] =$*
109        *$\nabla f_i(x)$, $\mathbb{E}[\|g_i(x) - \nabla f_i(x)\|_2^2] \leq \zeta^2$.*

110      *A.2 Each $f_i$ is smooth, i.e., for any $x, y \in \mathbb{R}^d$, there exists some non-negative constans $L_1, ..., L_d$*
111        *such that $f(y) - f(x) \leq \langle\nabla f(x), y - x\rangle + \frac{\sum_{j=1}^d L_j(y(j)-x(j))^2}{2}$.*

112    *A.3 There exists some constant $f^*$ such that $f(x) \geq f^*, \forall x \in \mathbb{R}^d$*

113    *A.4 There exists a constant $G \geq 0$ such that $\|\nabla f_i(x)\| \leq G, \forall i = 1, \ldots, n$, and $x \in \mathbb{R}^d$.*

114  For the convergence rate analysis, we consider two cases, namely, the case with $z < +\infty$ and the
115  case of $z = \infty$.

### 3.1  Case 1: $z < +\infty$

117  As we can see from Lemma 1, there always exists some gradient bias when $z < +\infty$. In order to
118  bound it, we further assume that a higher order moment of the minibatch gradient noise is bounded.

119  **Assumption 2.** *There exists a constant $Q_z \geq 0$ such that for any $x \in \mathbb{R}^d$, we have*

$$\mathbb{E}[\|g_i(x) - \nabla f_i(x)\|_{4z+2}^{4z+2}] \leq Q_z. \tag{4}$$

120  **Theorem 1.** *Suppose that Assumption 1 and 2 hold. Denote $\bar{x}_{t,s} = \frac{1}{n} \sum_{i=1}^n x_{t,s}^i$ and $L_{\max} =$*
121  *$\max_j L_j$. Then, for $\eta = \eta_z \sigma$ and $\gamma \leq \frac{1}{L_{\max}}$, we have*

$$\mathbb{E}\left[\frac{1}{TE} \sum_{t=1}^T \sum_{s=1}^E \|\nabla f(\bar{x}_{t-1,s-1})\|^2\right] \leq \underbrace{\frac{2\mathbb{E}[f(x_0) - f^*]}{TE\gamma} + \frac{\gamma \zeta^2 L_{\max}}{n} + \frac{(E-1)(2E-1)\gamma^2 L_{\max}^2 G^2}{3}}_{\text{(a). Standard terms in FedAvg}}$$

$$\tag{5a}$$

$$+ \underbrace{\frac{2^{2z} E^{2z} \sqrt{Q_z + G^{4z+2}} G}{(2z+1)\sigma^{2z}} + \frac{\gamma 2^{4z} E^{4z+1}(Q_z + G^{4z+2}) L_{\max}}{2(2z+1)^2 \sigma^{4z}}}_{\text{(b). Bias terms}}$$

$$\tag{5b}$$

$$+ \underbrace{\frac{4\eta_z^2 \gamma \sigma^2 \sum_{j=1}^d L_j}{En}}_{\text{(c). Variance term}}. \tag{5c}$$

122  **When is the bound non-trivial?** Since we assume that the $\ell_2$-norm of gradient is bounded by $G$, all
123  the terms in the RHS of (5) should be no larger than $G^2$. For example, one can check that to have the
124  fisrt term in (5b) less than $G^2$, one requires $\sigma$ to be greater than $2E \left(Q_z + G^{4z}\right)^{\frac{1}{4z}}/(2z+1)^{\frac{1}{2z}}$.

125  **Bias-variance trade-off.** An interesting observation from Theorem 1 is that there exists a trade-off
126  between the bias and variance terms. One can see that the terms in (5b) is caused by the gradient bias
127  of the sign operation (see (1)) and is an infinitesimal of $\sigma$ with $\mathcal{O}\left(\sigma^{-2z}\right)$, while the term in (5c) is due
128  to the injected noise and is in the order of $\mathcal{O}\left(\gamma\sigma^2\right)$. Specifically, the first term in (b) only depends on
129  the noise scale $\sigma$ and mostly affects the final learning performance. In the meanwhile, the variance
130  term mainly affects the convergence speed because a smaller stepsize is required for it to diminish.

131  Theoretically, we can choose an iteration-dependent noise scale $\sigma$ so as to making the algorithm
132  converge. In the following results, we denote $\tau = TE$ as the total number of gradient queries to the
133  local objective function.

134  **Corollary 1** (Informal). *Let $\gamma = \min\{n^{\frac{z}{2z+1}} \tau^{-\frac{z+1}{2z+1}}, \frac{1}{L_{\max}}\}$ and $\sigma = (n\tau)^{\frac{1}{4z+2}}$ in Theorem 1, and*
135  *let $E \leq n^{-\frac{3z}{4z+2}} \tau^{\frac{z+2}{4z+2}}$. We have*

$$\mathbb{E}\left[\frac{1}{\tau} \sum_{t=1}^T \sum_{s=1}^E \|\nabla f(\bar{x}_{t-1,s-1})\|^2\right] = \mathcal{O}\left((n\tau)^{-\frac{z}{2z+1}}\right). \tag{6}$$

136  **Achieveing linear speedup.** From Corollary 1, we can see that the $z$-SignFedAvg needs $(n\tau)^{\frac{3z}{4z+2}}$
137  communication rounds to achieve a linear speedup convergence rate. Particularly, when $z = 1$, the
138  corresponding convergence speed is $\mathcal{O}((n\tau)^{-\frac{1}{3}})$ and the required communication rounds is $(n\tau)^{\frac{1}{2}}$.

### 3.2  Case 2: $z = +\infty$

140  In this case, the injected noise in $z$-SignFedAvg is uniformly distributed at $[-1, 1]$. From Remark 1
141  we have learned that the bias term in (5b) will either blow up when $\sigma$ is lower than some threshold, or
142  vanish on the other hand. To quantify this threshold, we need the limit form of Assumption 2:

**Assumption 3.** *There exists a constant $Q_\infty \geq 0$ such that, with probability 1 we have*

$$\|g_i(x) - \nabla f_i(x)\|_\infty \leq Q_\infty, \forall x \in \mathbb{R}^d. \tag{7}$$

**Theorem 2** (Informal)**.** *Suppose that Assumption 1 and 3 hold. For $\gamma = \min\{n^{\frac{1}{2}}\tau^{-\frac{1}{2}}, \frac{1}{L_{\max}}\}$, $\eta = \sigma$, $E \leq n^{-\frac{3}{4}}\tau^{\frac{1}{4}}$, and $\sigma > E(G + Q_\infty)$, we have*

$$\mathbb{E}\left[\frac{1}{\tau}\sum_{t=1}^{T}\sum_{s=1}^{E}\|\nabla f(\bar{x}_{t-1,s-1})\|^2\right] = \mathcal{O}\left((n\tau)^{-\frac{1}{2}}\right). \tag{8}$$

We can see that (8) implies $\infty$-SignFedAvg recovers the convergence rate of uncompressed algorithms [Yu et al., 2019].

**Remark 2.** *It is worthwhile to point out that the condition of sufficeintly large noise scale $\sigma > E(G + Q_\infty)$ is necessary and cannot be spared. By intuition, this is because when $\sigma \leq E(G + Q_\infty)$ in Theorem 2, the injected uniform noise cannot change the sign of gradients in the worst case. For example, if $\xi_\infty$ follows uniform distribution on $[-1, 1]$, and now $\sigma < A$ for some $A > 0$, we have $\text{Sign}(x + \sigma\xi_\infty) = \text{Sign}(x)$ for any $x \geq A$.*

**Remark 3.** *By comparing the required thresholds for $\sigma$ in Theorem 1 and Theorem 2, we can see that when there is no minibatch gradient noise (i.e., $\zeta = 0$), Case 2 demands less noise injection and may perform better than Case 1. On the contrary, when the minibatch gradient noise has a long tail such as $Q_z \ll Q_\infty^{4z}$, Case 1 may be better. Despite of the distinction in theory, as we will see in Section 4, Case 1 and Case 2 have almost the same behavior on real datasets.*

More detailed theoretical results and comparsion with existing methods are relegated to Appendix B.

# 4 Experiments

In this section, we present the experiment results on real datasets. All the figures are obtained by 10 independent runs and are visualized in the form of mean±std. We also conduct an experiment on synthetic data where there is no minibatch gradient noise, and the results is relegated to Appendix D.

**Noise scale as a hyperparameter.** Although we explicitly characterize how the performance of Algorithm 1 depends on the noise scale $\sigma$ in previous section, we treat $\sigma$ as a tunable hyperparameter in practice. Because, on one hand, it usually impossible to compute the moment and support of the gradient noise in reality. One the other hand, since the theoretical results above only provide a worst-case guarantee, for some real problems, the optimal noise scales selected from grid search can be much smaller than the choice suggested by theory.

## 4.1 An extremely non-i.i.d setting

In this section, we consider an extremely non-i.i.d setting with the well-known dataset MNIST [Deng, 2012] which is a hand-written digits recogonition dataset. Specifically, we split the dataset into 10 parts based on the labels and each client only have the data of one digit. In such a highly heterogeneous setting, there is no benefit from local computation with periodic commnunication. Therefore, we compare with the listed algorithms: SGDwM [Ghadimi and Lan, 2013], EF-SignSGDwM [Karimireddy et al., 2019, Vogels et al., 2019], Sto-SignSGDwM [Safaryan and Richtárik, 2021]. Some baseline algorithms have an additional hyperparameter for the momentum of gradient. For all the algorithms, we select the their own optimal hyperparameters like stepsize, momentum coefficient, noise scale via grid search. For more details like hyperparameters for all the tested algorithms and the performance of 1-SignSGD and $\infty$-SignSGD under different noise scales, we refer the readers to Appendix E.1.

**Results.** From Figure 1a, 1b, we can observe that 1-SignSGD and $\infty$-SignSGD have roughly the same performance which outperform other sign-based algorithms and is slightly inferior to the uncompressed algorithm. Our theory is verified by comparing the performance of noiseless SignSGD and our proposed algortihms. If we compare the performance with respect to the total number of transmitted bits, our algorithms achieve the state-of-the-art performance on this task as we can see in Figure 1c.

## 4.2 Federated Learning on EMNIST

In this section, we verify the performance of our proposed Algorithm 1 on EMNIST[Cohen et al., 2017]. We mainly compare the performance of three algorithms: FedAvg without any compression [McMahan et al., 2017, Yu et al., 2019] and our proposed Algorithm 1 with $z = 1$ and $z = \infty$.

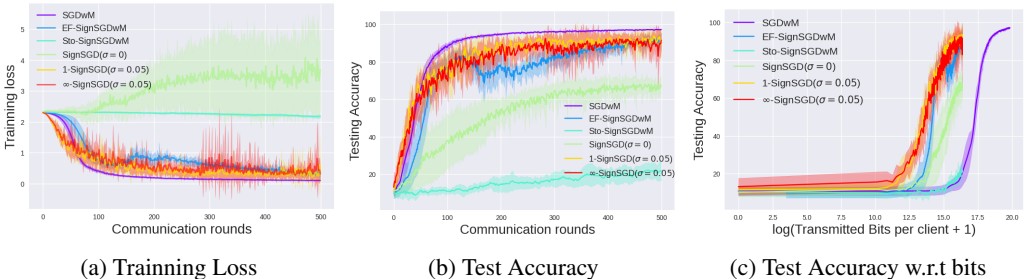

| (a) Trainning Loss | (b) Test Accuracy | (c) Test Accuracy w.r.t bits |

Figure 1: Performance of various algorithms on non-i.i.d MNIST

**Settings.** We follow a similar setting to [Reddi et al., 2020]. We also consider the scenario with partial participation. Specifically, for the EMNIST dataset, there are 3579 clients in total and we sample 100 clients uniformly to upload their local gradients at each communication round.

**Results.** The hyperparameters for the algorithms are tuned via grid search and details are in Appendix E.2. Specifically, we use $\sigma = 0.01$ for both 1-SignFedAvg and $\infty$-SignFedAvg on EMNIST dataset. We can see from Figure 2 that all the algorithms can benefit from multiple local steps, and more surprisingly, both 1-SignFedAvg and $\infty$-SignFedAvg can outperform the umcompressed algorithm FedAvg. This is probably because the EMNIST dataset is less non-i.i.d as the dataset we use in Section 4.1. The performance of 1-SignFedAvg and $\infty$-SignFedAvg under various choices of noise scale are relegated to the Figure 6 and 7 in Appendix E.2, which also matches our theoretical results.

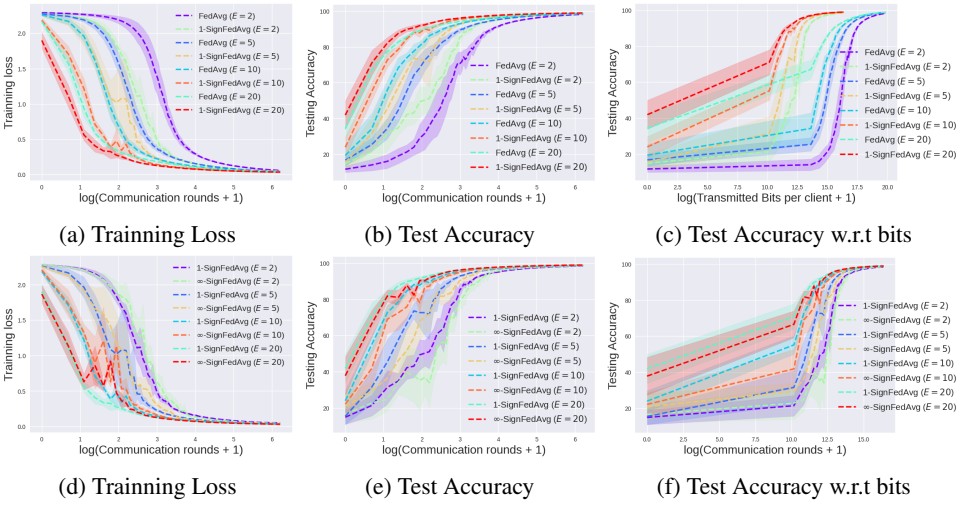

| (a) Trainning Loss | (b) Test Accuracy | (c) Test Accuracy w.r.t bits |
| (d) Trainning Loss | (e) Test Accuracy | (f) Test Accuracy w.r.t bits |

Figure 2: Performance of FedAvg, 1-SignFedAvg and $\infty$-SignFedAvg on EMNIST dataset.

## 5 Conclusion

In this work, we have proposed the $z$-SignFedAvg: a FedAvg-type algorithm with a novel family of sign-based stochastic compression. Throughout extensive theoretical analysis and empirical evaluation, we have shown that $z$-SignFedAvg can perform comparably, sometimes even better, as the uncompressed FedAvg algorithm with a significantly reduced number of bits transmitted from the clients to the server. However, a vital issue in $z$-SignFedAvg is that it involves a new hyperparameter, i.e., the noise scale $\sigma$, which needs to be carefully chosen for achieving a good convergence performance. An interesting observation from the experiments is that the less heterogeneous the local data are, the smaller the optimal noise scale is, which is consisten with the theoretical insights. In the future, we will futher study the relationship between the client's heterogeneity and the optimal noise scale. As a final remark, we note that the stochastic sign-based compression proposed in this work is of independent interest and can be directly combined with other adaptive FL algorithms like those in Karimireddy et al. [2020], Reddi et al. [2020].

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

# Appendices

## A  Dicussion on related works

Sign-based optimization algorithms such as the SignSGD in [Bernstein et al., 2018] have gained much popularity recently because of their simple compression rule and comparable performance to uncompressed algorithms. In this work, we focus on the scenario with heterogeneous data, and as we have discussed in Section 1, a naive application of sign-based compression in this scenario is problematic. Besides, we consider using sign-based compression only for the uplink communication in this work, while it is worth mentioning that [Tang et al., 2019, Jin et al., 2020, Chen et al., 2020a] also compress for the downlink communication. In the following paragraphs, we review a few related works on similar topics.

**Stochastic sign-based method.**  The setting considered by [Safaryan and Richtárik, 2021] is the closest to ours because [Jin et al., 2020, Chen et al., 2020a] also compresses the server-to-client communication with majority vote. Aside from the difference in setting, the algorithms in them achieve the same convergence rate $O(\tau^{-\frac{1}{4}})$ w.r.t different convergence metrics, where $\tau$ is the number of gradient queries to the objective function. As will be discussed in Appendix A, these rates are usually inferior to that of uncompressed algorithms. Our proposed algorithm also belongs to this category. Compared to existing works, we require a slightly stronger assumption on the gradient noise, and the convergence speed of our algorithm is either $O(\tau^{-\frac{1}{3}})$ or $O(\tau^{-\frac{1}{2}})$ with the commonly used squared $\ell_2$ norm of gradients as the convergence metric. Moreover, we also show that our proposed sign-based algorithm can achieve a linear speedup when the number of clients increases, and such a result is not known in previous works.

**Error Feedback method.** The error feedback (EF) method is first proposed by [Seide et al., 2014] and then theoretically justified by [Karimireddy et al., 2019]. Then [Vogels et al., 2019, Tang et al., 2019, 2021b] further extend this EF framework into distributed non-i.i.d setting and adaptive gradient method. The key idea is to show that the sign operator multiplying with one norm is a contractive compressor, and the error induced by the contractive compressor can be fixed by the error-compensated gradient method. However, unlike the pure sign-based gradient method, it must transmit one extra real number for the one norm. Besides, the convergence rate for the EF algorithms is $\mathcal{O}(\tau^{-\frac{1}{2}} + \tau^{-1}/\delta^2)$, where $\delta$ is the parameter of contractive compressor. In the worst case, the sign operator multiplying with one norm is a contractive compressor with $\delta = 1/d$, where $d$ is the dimension of the gradient. Therefore, the convergence rate of it becomes $\mathcal{O}(\tau^{-\frac{1}{2}} + d^2\tau^{-1})$, which could become very bad especially for high-dimension optimization problem. Besides, to our knowledge, no one has extended the error feedback method to the scenario with periodic aggregation.

It is often tricky to compare the convergence results of sign-based methods because some works like [Chen et al., 2020a, Safaryan and Richtárik, 2021] do not use the standard convergence metric. To better compare existing results and ours, in Appendix A, we provide a detailed discussion on the existing convergence metrics and summarize the representative algorithms and their theoretical results in Table 1.

Table 1 gives a brief summary for a few representative works related to this work. In this table, we review the algorithms in these works on the convergence rate along with the used convergence metrics, communication complexity, assumptions required, and also whether they can deal with periodic aggregation. Particularly, [Chen et al., 2020a, Safaryan and Richtárik, 2021] adopt convergence metrics other than the commonly used average squared $\ell_2$ norm of gradients. Due to the additional step of server-to-client compression, [Chen et al., 2020a] use a convergence metric mixed with $\ell_2$ norm and $\ell_1$ norm, while [Safaryan and Richtárik, 2021] use $\ell_2$ norm instead of squared $\ell_2$ norm. For communication complexity, we only compare the unlink communication, and to compute the used bits per communication, we assume that all the algorithms need 32 bits to represent a float number as this is the most common setting in Tensorflow [Abadi et al., 2016b] and Pytorch [Paszke et al., 2017].

Among the works in Table 1, the setting considered by [Safaryan and Richtárik, 2021] is the closest to ours. [Safaryan and Richtárik, 2021] propose an algorithm that can achieve convergence rate $\mathcal{O}(\tau^{-\frac{1}{4}})$ with average $\ell_2$ norm of gradients as the metric. We remark that this is inferior to the convergence rate $\mathcal{O}(\tau^{-\frac{1}{2}})$ with squared $\ell_2$ norm as the metric. To illustrate this point, we denote a serie of vector

| Algorithm | Convergence metric / rate | Used bits per communication | Extra Assumptions? | Can achieve linear speedup? | Can allow multiple local steps? |
|---|---|---|---|---|---|
| [Ghadimi and Lan, 2013] | $\mathcal{O}(\tau^{-\frac{1}{2}})$ squared $\ell_2$ | $32d$ | No | ✓ | ✗ |
| [Yu et al., 2019] | $\mathcal{O}(\tau^{-\frac{1}{2}})$ squared $\ell_2$ | $32d$ | • Bounded gradient | ✓ | ✓ |
| [Karimireddy et al., 2019] | $\mathcal{O}(\tau^{-\frac{1}{2}} + d^2\tau^{-1})$ squared $\ell_2$ | $d + 32$ | • Bounded gradient | ✗ | ✗ |
| [Safaryan and Richtárik, 2021] | $\mathcal{O}(\tau^{-\frac{1}{4}})$ $\ell_2$ | $d$ | No | ✗ | ✗ |
| [Jin et al., 2020] | $\mathcal{O}(\tau^{-\frac{1}{4}})$ squared $\ell_2$ | $d$ | • Bounded gradient • n is an odd number | ✗ | ✗ |
| [Chen et al., 2020a] | $\mathcal{O}(\tau^{-\frac{1}{4}})$ mixed | $d$ | • Bounded gradient • n is an odd number | ✗ | ✗ |
| 1-SignFedAvg (ALG. 1) **This work** | $\mathcal{O}(\tau^{-\frac{1}{3}})$ squared $\ell_2$ | $d$ | • Bounded gradient • Bounded 6th moment of gradient noise | ✓ | ✓ |
| $\infty$-SignFedAvg (ALG. 1) **This work** | $\mathcal{O}(\tau^{-\frac{1}{2}})$ squared $\ell_2$ | $d$ | • Bounded gradient • Bounded support of gradient noise | ✓ | ✓ |

Table 1: Summary for related works.

453    $\{\alpha_1, ..., \alpha_\tau, ...\}$ with $\alpha_i \in \mathbb{R}^d$. If now

$$\frac{1}{\tau}\sum_{i=1}^{\tau}\|\alpha_i\| = \mathcal{O}(\tau^{-\frac{1}{4}}), \tag{9}$$

454    in the worst case, we can only guarantee that

$$\frac{1}{\tau}\sum_{i=1}^{\tau}\|\alpha_i\|^2 \le \tau\left(\frac{1}{\tau}\sum_{i=1}^{\tau}\|\alpha_i\|\right)^2 = \mathcal{O}(\tau^{\frac{1}{2}}) \tag{10}$$

455    for squared $\ell_2$ norm. As a simple example, the equality in (10) holds when there is only one non-zero
456    term in $\{\alpha_1, ..., \alpha_\tau\}$.

457    On the contrary, if

$$\frac{1}{\tau}\sum_{i=1}^{\tau}\|\alpha_i\|^2 = \mathcal{O}(\tau^{-\frac{1}{2}}), \tag{11}$$

458    we have

$$\frac{1}{\tau}\sum_{i=1}^{\tau}\|\alpha_i\| \le \sqrt{\frac{1}{\tau}\sum_{i=1}^{\tau}\|\alpha_i\|^2} = \mathcal{O}(\tau^{-\frac{1}{4}}). \tag{12}$$

459    Consider the scenario $E = 1$, the algorithm in [Safaryan and Richtárik, 2021] is equivalent to our
460    Algorithm 1 with $\sigma$ chosen to be $\|g_{t-1,s}^i\|$. On one hand, this choice of noise scale $\sigma$ make it unable
461    to be extended to the federated averaging algorithm, because each client use a different noise scale.
462    On the other hand, this choice is linearly increaseing w.r.t problem dimension and hence is too
463    conservative. From Figure 3 and 1 we can see that this input-dependent noise scale result in an
464    extremely slow convergence for high-dimension problems.

465    **B   Theoretical results**

466    In this section, we state the formal version of Corollary 1 and Theorem 2.

467 **Corollary 2** (Formal version of Corollary 1)**.** *If we choose* $\gamma = \min\{n^{\frac{z}{2z+1}}\tau^{-\frac{z+1}{2z+1}}, \frac{1}{L_{\max}}\}$ *and*

468 $\sigma = (n\tau)^{\frac{1}{4z+2}}$ *in Theorem 1, we have*

$$
\mathbb{E}[\frac{1}{\tau}\sum_{t=1}^{T}\sum_{s=1}^{E}\|\nabla f(\bar{x}_{t-1,s-1})\|^2 \leq \frac{2\mathbb{E}[f(x_0)-f^*]}{(n\tau)^{\frac{z}{2z+1}}} + \frac{\zeta^2 L_{\max}}{(n\tau)^{\frac{z+1}{2z+1}}} + \frac{(E-1)(2E-1)n^{\frac{2z}{2z+1}}L_{\max}^2 G^2}{3\tau^{\frac{2z+2}{2z+1}}}
$$

(13a)

$$
+ \frac{2^{2z}E^{2z}\sqrt{Q_z + G^{4z+2}}G}{(2z+1)(n\tau)^{\frac{z}{2z+1}}} + \frac{2^{4z}E^{4z+1}(Q_z + G^{4z+2})L_{\max}}{2(2z+1)^2 n^{\frac{z}{2z+1}}\tau^{\frac{3z+1}{2z+1}}}
$$

(13b)

$$
+ \frac{4\eta_z^2 \sum_{j=1}^{d}L_j}{E(n\tau)^{\frac{z}{2z+1}}}.
$$

(13c)

469 *Furthermore, if $E \leq n^{-\frac{3z}{4z+2}}\tau^{\frac{z+2}{4z+2}}$, the upper bound above will converge as $\mathcal{O}\left((n\tau)^{-\frac{z}{2z+1}}\right)$.*

470 **Relationship to [Chen et al., 2020a].** [Chen et al., 2020a] also studies the sign-based optimization
471 algorithm with symmetric and zero-mean noise and prove that the convergence rate is $\mathcal{O}(\tau^{-\frac{1}{4}})$ using
472 a similar iteration-dependent noise scale like us. However, there are two difference between their
473 result and our Theorem 1. First, since they also consider the downlink compression, the convergence
474 metric they used is no longer $\ell_2$ norm and hard to interpret. Second, unlike [Chen et al., 2020a]
475 whose result is rooted in median-based algorithm, our analysis directly exploits the property of sign
476 operation and hence can provide a better and more interpretable result.

477 **Theorem 3** (Formal version of Theorem 2)**.** *Given that Assumption 1 and 3 hold, and we choose*
478 $\eta = \sigma$, *if $\gamma \leq \frac{1}{L_{\max}}$, if $\sigma > E(G+Q_\infty)$, we have*

$$
\mathbb{E}[\frac{1}{TE}\sum_{t=1}^{T}\sum_{s=1}^{E}\|\nabla f(\bar{x}_{t-1,s-1})\|^2 \leq \underbrace{\frac{2\mathbb{E}[f(x_0)-f^*]}{TE\gamma} + \frac{\gamma\zeta^2 L_{\max}}{n} + \frac{(E-1)(2E-1)\gamma^2 L_{\max}^2 G^2}{3}}_{\text{standard terms in federated averaging}}
$$

(14a)

$$
+ \underbrace{\frac{4\gamma\sigma^2 \sum_{j=1}^{d}L_j}{En}}_{\text{variance term}}.
$$

(14b)

479 *otherwise, if $\sigma \leq E(G+Q_\infty)$, there exists a problem where the algorithm cannot converge.*

480 *If we further choose $\gamma = \min\{n^{\frac{1}{2}}\tau^{-\frac{1}{2}}, \frac{1}{L_{\max}}\}$, we have*

$$
\mathbb{E}[\frac{1}{\tau}\sum_{t=1}^{T}\sum_{s=1}^{E}\|\nabla f(\bar{x}_{t-1,s-1})\|^2 \leq \frac{2\mathbb{E}[f(x_0)-f^*]}{(n\tau)^{\frac{1}{2}}} + \frac{\zeta^2 L_{\max}}{(n\tau)^{\frac{1}{2}}} + \frac{(E-1)(2E-1)nL_{\max}^2 G^2}{3\tau}
$$

(15)

$$
+ \frac{4\sigma^2 \sum_{j=1}^{d}L_j}{E(n\tau)^{\frac{1}{2}}}.
$$

(16)

481 *Furthermore, if $E \leq n^{-\frac{3}{4}}\tau^{\frac{1}{4}}$, the upper bound above will converge as $\mathcal{O}\left((n\tau)^{-\frac{1}{2}}\right)$, which recovers*
482 *the convergence result of uncompressed algorithm [Yu et al., 2019].*

483 **Remark 4.** *When $\sigma \leq E(G+Q_\infty)$ in Theorem 3, the injected uniform noise cannot change the sign*
484 *of gradients in the worst case. For example, if $\xi_\infty$ follows uniform distribution on $[-1,1]$, and now*
485 *$\sigma < A$ for some $A > 0$, we have $\text{Sign}(x + \sigma\xi_\infty) = \text{Sign}(x)$ for any $x \geq A$.*

486 **Relationship to [Jin et al., 2020, Safaryan and Richtárik, 2021].** We remark that both the
487 stochastic sign operators proposed in [Jin et al., 2020, Safaryan and Richtárik, 2021] are equivalent
488 to the sign operator with uniform noise considered in Case 2. In particular, [Jin et al., 2020] also

consider downlink compression and hence its convergence results are not directly comparable to the Case 2. [Safaryan and Richtárik, 2021] adopts an input-dependent noise scale and proves $\mathcal{O}(\tau^{-\frac{1}{4}})$ convergence rate with $\ell_2$ norm of gradient as the metric. We remark that this rate is usually worse than the rate $\mathcal{O}(\tau^{-\frac{1}{2}})$ with squared $\ell_2$ norm as the metric. Such input-dependent noise scale make it possible to prove convergence without the bounded support of gradient noise assumed in this work. But there are two disadvantages for their choice of noise scale. First, it can not be extended to federated averaging algorithm. Second, it often leads to slow convergence in practice when the problem dimension is very high. More discussions on Safaryan and Richtárik [2021] are provided in Appendix A.

## C  Missing proofs

**Lemma 2.** *z-distribution weakly converges to uniform distribution at $[-1, 1]$ when $z \to +\infty$.*

*Proof of Lemma 2.* Now we denote the p.d.f of uniform distribution as

$$p_\infty(x) = \begin{cases} \frac{1}{2} & |x| \leq 1, \\ 0 & |x| > 1. \end{cases} \tag{17}$$

Without loss of generality, for any $x > 1$ and $z \in \mathbb{Z}_+$, we have

$$\left| \int_{-\infty}^{x} \frac{1}{2\eta_z} e^{-\frac{t^{2z}}{2}} dt - \int_{-\infty}^{x} p_\infty(t) dt \right| = \left| \int_{0}^{x} \left( \frac{1}{2\eta_z} e^{-\frac{t^{2z}}{2}} - p_\infty(t) \right) dt \right| \tag{18a}$$

$$\leq \int_{0}^{1} \left| \frac{1}{2\eta_z} e^{-\frac{t^{2z}}{2}} - \frac{1}{2} \right| dt + \int_{1}^{x} \frac{1}{2\eta_z} e^{-\frac{t^{2z}}{2}} dt. \tag{18b}$$

For any $0 < \epsilon < \min\{1, x - 1\}$, we have

$$\int_{0}^{1} \left| \frac{1}{2\eta_z} e^{-\frac{t^{2z}}{2}} - \frac{1}{2} \right| dt = \int_{0}^{1-\epsilon} \left| \frac{1}{2\eta_z} e^{-\frac{t^{2z}}{2}} - \frac{1}{2} \right| dt + \int_{1-\epsilon}^{1} \left| \frac{1}{2\eta_z} e^{-\frac{t^{2z}}{2}} - \frac{1}{2} \right| dt \tag{19a}$$

$$\leq \left| \frac{1}{2\eta_z} e^{-\frac{(1-\epsilon)^{2z}}{2}} - \frac{1}{2} \right| + \epsilon. \tag{19b}$$

Since $\lim_{z\to\infty} \frac{1}{2\eta_z} = \lim_{z\to\infty} \frac{z}{2^{\frac{1}{2z}} \Gamma(\frac{1}{2z})} = \frac{1}{2}$ and $\lim_{z\to\infty} e^{-\frac{(1-\epsilon)^{2z}}{2}} = 1$, there exists an interger $Z_1 > 0$ such that if $z > Z_1$, we have

$$\left| \frac{1}{2\eta_z} e^{-\frac{(1-\epsilon)^{2z}}{2}} - \frac{1}{2} \right| \leq \epsilon.$$

Similarly, we have

$$\int_{1}^{x} \frac{1}{2\eta_z} e^{-\frac{t^{2z}}{2}} dt = \int_{1}^{1+\epsilon} \frac{1}{2\eta_z} e^{-\frac{t^{2z}}{2}} dt + \int_{1+\epsilon}^{x} \frac{1}{2\eta_z} e^{-\frac{t^{2z}}{2}} dt \tag{20a}$$

$$\leq \epsilon + \frac{1}{2\eta_z} e^{-\frac{(1+\epsilon)^{2z}}{2}} (x - 1 - \epsilon). \tag{20b}$$

Since $\lim_{z\to\infty} e^{-\frac{(1+\epsilon)^{2z}}{2}} = 0$, there exists an interger $Z_2 > 0$ such that if $z > Z_2$, we have

$$\int_{1}^{x} \frac{1}{2\eta_z} e^{-\frac{t^{2z}}{2}} dt \leq \epsilon. \tag{21}$$

In all, for any $0 < \epsilon < 1$, if $z$ is sufficiently large, we have

$$\left| \int_{-\infty}^{x} \frac{1}{2\eta_z} e^{-\frac{t^{2z}}{2}} dt - \int_{-\infty}^{x} p_\infty(t) dt \right| \leq 4\epsilon. \tag{22}$$

Take $\epsilon \to 0$ and $z \to \infty$, we have

$$\lim_{z\to\infty}\left|\int_{-\infty}^{x}\frac{1}{2\eta_z}e^{-\frac{t^{2z}}{2}}dt - \int_{-\infty}^{x}p_\infty(t)dt\right| = 0. \tag{23}$$

$\square$

*Proof of Lemma 1.* We first state a useful inequality on the c.d.f of $z$ distribution:

**Lemma 3.** *For any $x \in \mathbb{R}$*

$$|x| - \frac{|x|^{2z+1}}{2(2z+1)} \leq |\Psi_z(x)| \leq |x|, \text{ where } \Psi_z(x) \stackrel{\text{def.}}{=} \int_0^x e^{-\frac{t^{2z}}{2}}dt. \tag{24}$$

Then, we have

$$\|\eta_z\sigma\mathbb{E}\left[\text{Sign}(\mathbf{x}+\sigma\xi_\mathbf{z})\right] - x\|^2 = \left\|x - \sigma\Psi_z(\frac{x}{\sigma})\right\|^2 = \sum_{j=1}^{d}\left(x(j) - \sigma\Psi_z(\frac{x(j)}{\sigma})\right)^2 \tag{25a}$$

$$\leq \sum_{j=1}^{d}\frac{x(j)^{4z+2}}{4(2z+1)^2\sigma^{4z}} = \frac{\|x\|_{4z+2}^{4z+2}}{4(2z+1)^2\sigma^{4z}}. \tag{25b}$$

$\square$

*Proof of Lemma 3.* Without loss of generality, we prove it for $x \geq 0$.

First,

$$\int_0^x e^{-\frac{t^{2z}}{2}}dt \leq \int_0^x 1dt \leq x. \tag{26}$$

Now we define $F(x) \stackrel{\text{def.}}{=} \int_0^x e^{-\frac{t^{2z}}{2}}dt - x + \frac{x^{2z+1}}{2(2z+1)}$. Note that $F(0) = 0$.

Then, we can prove that $F(x) \geq 0$ by

$$F'(x) = e^{-\frac{t^{2z}}{2}} - x + \frac{t^{2z}}{2} \geq 0. \tag{27}$$

(27) is due to the inequality $e^{-x} - 1 + x \geq 0$ for any $x \geq 0$. $\square$

*Proof of Theorem 1.* Here we define the virtual aggregated update:

$$\bar{x}_{t,s} = \frac{1}{n}\sum_{i=1}^{n}x_{t,s}^i, \tag{28}$$

$$\bar{x}_t = \bar{x}_{t-1,E}. \tag{29}$$

We now state the two useful lemmas:

**Lemma 4.**

$$\mathbb{E}[f(x_t) - f(\bar{x}_t)] \leq \frac{\gamma 2^{2z}E^{2z+1}\sqrt{Q_z + G^{4z+2}}G}{2(2z+1)\sigma^{2z}} + \frac{\gamma^2 2^{4z}E^{4z+2}(Q_z + G^{4z+2})L_{\max}}{4(2z+1)^2\sigma^{4z}} \tag{30a}$$

$$+ \frac{2\eta_z^2\gamma^2\sigma^2\sum_{j=1}^{d}L_j}{n}. \tag{30b}$$

**Lemma 5.**

$$\mathbb{E}[f(\bar{x}_t) - f(x_{t-1})] \leq -\frac{\gamma}{2}\sum_{s=1}^{E}\|\nabla f(\bar{x}_{t-1,s-1})\|^2 + \frac{E\gamma^2\zeta^2L_{\max}}{2n} + \frac{E(E-1)(2E-1)\gamma^3L_{\max}^2G^2}{6}. \tag{31}$$

With this two lemma, we have

$$\mathbb{E}[f(x_t) - f(x_{t-1})] = \mathbb{E}[f(x_t) - f(\bar{x}_t)] + E[f(\bar{x}_t) - f(x_{t-1})] \tag{32a}$$

$$\leq -\frac{\gamma}{2} \sum_{s=1}^{E} \|\nabla f(\bar{x}_{t-1,s-1})\|^2 + \frac{E\gamma^2\zeta^2 L_{\max}}{2n} + \frac{E(E-1)(2E-1)\gamma^3 L_{\max}^2 G^2}{6} \tag{32b}$$

$$+ \frac{\gamma 2^{2z} E^{2z+1}\sqrt{Q_z + G^{4z+2}}G}{2(2z+1)\sigma^{2z}} + \frac{\gamma^2 2^{4z} E^{4z+2}(Q_z + G^{4z+2})L_{\max}}{4(2z+1)^2\sigma^{4z}} \tag{32c}$$

$$+ \frac{2\eta_z^2\gamma^2\sigma^2 \sum_{j=1}^{d} L_j}{n}. \tag{32d}$$

Rearranging the terms, we have

$$\frac{1}{E} \sum_{s=1}^{E} \|\nabla f(\bar{x}_{t-1,s-1})\|^2 \leq \frac{2\mathbb{E}[f(x_{t-1}) - f(x_t)]}{E\gamma} + \frac{\gamma\zeta^2 L_{\max}}{n} + \frac{(E-1)(2E-1)\gamma^2 L_{\max}^2 G^2}{3} \tag{33a}$$

$$+ \frac{2^{2z} E^{2z}\sqrt{Q_z + G^{4z+2}}G}{(2z+1)\sigma^{2z}} + \frac{\gamma 2^{4z} E^{4z+1}(Q_z + G^{4z+2})L_{\max}}{2(2z+1)^2\sigma^{4z}} \tag{33b}$$

$$+ \frac{4\eta_z^2\gamma\sigma^2 \sum_{j=1}^{d} L_j}{En}. \tag{33c}$$

Form the telescopic sum

$$\mathbb{E}[\frac{1}{TE} \sum_{t=1}^{T} \sum_{s=1}^{E}]\|\nabla f(\bar{x}_{t-1,s-1})\|^2 \leq \frac{2\mathbb{E}[f(x_0) - f^*]}{TE\gamma} + \frac{\gamma\zeta^2 L_{\max}}{n} + \frac{(E-1)(2E-1)\gamma^2 L_{\max}^2 G^2}{3} \tag{34a}$$

$$+ \frac{2^{2z} E^{2z}\sqrt{Q_z + G^{4z+2}}G}{(2z+1)\sigma^{2z}} + \frac{\gamma 2^{4z} E^{4z+1}(Q_z + G^{4z+2})L_{\max}}{2(2z+1)^2\sigma^{4z}} \tag{34b}$$

$$+ \frac{4\eta_z^2\gamma\sigma^2 \sum_{j=1}^{d} L_j}{En}. \tag{34c}$$

$\square$

*Proof of Lemma 4.* Therefore, from smoothness we have,

$$f(x_t) - f(\bar{x}_t) \leq \langle \nabla f(\bar{x}_t), x_t - \bar{x}_t \rangle + \frac{\sum_{j=1}^{d} L_j \left(x_t(j) - \bar{x}_t(j)\right)^2}{2}. \tag{35}$$

The following equation and inequality can be checked, where the expectation is taken over the noise vector $\xi_z$,

$$x_t - \bar{x}_t = \frac{\gamma}{n} \sum_{i=1}^{n} \left(\eta_z\sigma\text{Sign}\left(\sum_{s=1}^{E} g_{t,s}^i + \sigma\xi_z\right) - \sum_{s=1}^{E} g_{t,s}^i\right), \tag{36}$$

$$\mathbb{E}[x_t - \bar{x}_t] = \frac{\gamma}{n} \sum_{i=1}^{n} \left(\sigma\Psi_z\left(\frac{1}{\sigma}\sum_{s=1}^{E} g_{t,s}^i\right) - \sum_{s=1}^{E} g_{t,s}^i\right). \tag{37}$$

For any $j = 1, ..., d$, we have

$$\mathbb{E}[(x_t(j) - \bar{x}_t(j))^2] \leq \frac{\gamma^2}{n^2} \left( \sum_{i=1}^{n} \left( \sigma \Psi_z \left( \frac{1}{\sigma} \sum_{s=1}^{E} g_{t,s}^i(j) \right) - \sum_{s=1}^{E} g_{t,s}^i(j) \right) \right)^2 \tag{38a}$$

$$+ \frac{\gamma^2}{n^2} \mathbb{E}\left[ \left( \sum_{i=1}^{n} \left( \eta_z \sigma \mathrm{Sign} \left( \sum_{s=1}^{E} g_{t,s}^i(j) + \sigma \xi_z \right) - \sigma \Psi_z \left( \frac{1}{\sigma} \sum_{s=1}^{E} g_{t,s}^i(j) \right) \right) \right)^2 \right] \tag{38b}$$

$$\leq \frac{\gamma^2}{n} \sum_{i=1}^{n} \left( \sigma \Psi_z \left( \frac{1}{\sigma} \sum_{s=1}^{E} g_{t,s}^i(j) \right) - \sum_{s=1}^{E} g_{t,s}^i(j) \right)^2 \tag{38c}$$

$$+ \frac{\gamma^2}{n^2} \sum_{i=1}^{n} \mathbb{E}\left[ \left( \eta_z \sigma \mathrm{Sign} \left( \sum_{s=1}^{E} g_{t,s}^i(j) + \sigma \xi_z \right) - \sigma \Psi_z \left( \frac{1}{\sigma} \sum_{s=1}^{E} g_{t,s}^i(j) \right) \right)^2 \right] \tag{38d}$$

$$\leq \frac{\gamma^2}{n} \sum_{i=1}^{n} \left( \sigma \Psi_z \left( \frac{1}{\sigma} \sum_{s=1}^{E} g_{t,s}^i(j) \right) - \sum_{s=1}^{E} g_{t,s}^i(j) \right)^2 \tag{38e}$$

$$+ \frac{2\gamma^2}{n^2} \sum_{i=1}^{n} \mathbb{E}\left[ \left( \eta_z \sigma \mathrm{Sign} \left( \sum_{s=1}^{E} g_{t,s}^i(j) + \sigma \xi_z \right) \right)^2 \right] \tag{38f}$$

$$+ \frac{2\gamma^2}{n^2} \sum_{i=1}^{n} \left( \sigma \Psi_z \left( \frac{1}{\sigma} \sum_{s=1}^{E} g_{t,s}^i(j) \right) \right)^2 \tag{38g}$$

$$\leq \frac{\gamma^2}{n} \sum_{i=1}^{n} \left( \sigma \Psi_z \left( \frac{1}{\sigma} \sum_{s=1}^{E} g_{t,s}^i(j) \right) - \sum_{s=1}^{E} g_{t,s}^i(j) \right)^2 + \frac{4\eta_z^2 \gamma^2 \sigma^2}{n}. \tag{38h}$$

$$\tag{38i}$$

Therefore, from Lemma 1, we have

$$\mathbb{E}[\sum_{j=1}^{d} L_j (x_t(j) - \bar{x}_t(j))^2] \leq \frac{\gamma^2}{n} \sum_{i=1}^{n} \sum_{j=1}^{d} L_j \left( \sigma \Psi_z \left( \frac{1}{\sigma} \sum_{s=1}^{E} g_{t,s}^i(j) \right) - \sum_{s=1}^{E} g_{t,s}^i(j) \right)^2 \tag{39a}$$

$$+ \frac{4\eta_z^2 \gamma^2 \sigma^2 \sum_{j=1}^{d} L_j}{n} \tag{39b}$$

$$\leq \frac{\gamma^2 L_{\max}}{n} \sum_{i=1}^{n} \left\| \sigma \Psi_z \left( \frac{1}{\sigma} \sum_{s=1}^{E} g_{t,s}^i \right) - \sum_{s=1}^{E} g_{t,s}^i \right\|^2 + \frac{4\eta_z^2 \gamma^2 \sigma^2 \sum_{j=1}^{d} L_j}{n} \tag{39c}$$

$$\leq \frac{\gamma^2 L_{\max}}{4(2z+1)^2 \sigma^{4z} n} \sum_{i=1}^{n} \left\| \sum_{s=1}^{E} g_{t,s}^i \right\|_{4z+2}^{4z+2} + \frac{4\eta_z^2 \gamma^2 \sigma^2 \sum_{j=1}^{d} L_j}{n}. \tag{39d}$$

Now we need to bound

$$\mathbb{E}\left[ \left\| \sum_{s=1}^{E} g_{t,s}^i \right\|_{4z+2}^{4z+2} \right], \tag{40}$$

where the expectation is taken over gradient noise.

$$\mathbb{E}\left[\left\|\sum_{s=1}^{E} g_{t,s}^i\right\|_{4z+2}^{4z+2}\right] \leq \mathbb{E}\left[E^{4z+1}\sum_{s=1}^{E}\left\|g_{t,s}^i\right\|_{4z+2}^{4z+2}\right] \tag{41a}$$

$$= \mathbb{E}\left[E^{4z+1}\sum_{s=1}^{E}\left\|g_{t,s}^i - \nabla f_i(x_{t,s-1}^i) + \nabla f_i(x_{t,s-1}^i)\right\|_{4z+2}^{4z+2}\right] \tag{41b}$$

$$\leq \mathbb{E}\left[(2E)^{4z+1}\sum_{s=1}^{E}\left\|g_{t,s}^i - \nabla f_i(x_{t,s-1}^i)\right\|_{4z+2}^{4z+2} + (2E)^{4z+1}\sum_{s=1}^{E}\left\|\nabla f_i(x_{t,s-1}^i)\right\|_{4z+2}^{4z+2}\right] \tag{41c}$$

$$\leq (2E)^{4z+1}EQ_z + (2E)^{4z+1}\sum_{s=1}^{E}\left\|\nabla f_i(x_{t,s-1}^i)\right\|_2^{4z+2} \leq 2^{4z+1}E^{4z+2}(Q_z + G^{4z+2}). \tag{41d}$$

In the derivation above, we use a classical result on the monotonicity of $\ell_p$ norm: For any $x \in \mathbb{R}^d$ and $1 < r < p$, we have

$$\|x\|_p \leq \|x\|_r \leq d^{\frac{1}{r}-\frac{1}{p}}\|x\|_p. \tag{42}$$

Therefore, by taking expectation over both $\xi_z$ and Gradient noise, we have

$$\mathbb{E}[\sum_{j=1}^{d} L_j\left(x_t(j) - \bar{x}_t(j)\right)^2] \leq \frac{\gamma^2 2^{4z+1}E^{4z+2}(Q_z + G^{4z+2})L_{\max}}{4(2z+1)^2\sigma^{4z}} + \frac{4\eta_z^2\gamma^2\sigma^2\sum_{j=1}^{d}L_j}{n}. \tag{43}$$

Hence, we have

$$\mathbb{E}[f(x_t) - f(\bar{x}_t)] \leq \left\langle \nabla f(\bar{x}_t), \frac{\gamma}{n}\sum_{i=1}^{n}\left(\sigma\Psi\left(\frac{1}{\sigma}\sum_{s=1}^{E}g_{t,s}^i\right) - \sum_{s=1}^{E}g_{t,s}^i\right)\right\rangle + \frac{\sum_{j=1}^{d}L_j\left(x_t(j) - \bar{x}_t(j)\right)^2}{2} \tag{44a}$$

$$\leq \|\nabla f(\bar{x}_t)\|\left\|\frac{\gamma}{n}\sum_{i=1}^{n}\left(\sigma\Psi\left(\frac{1}{\sigma}\sum_{s=1}^{E}g_{t,s}^i\right) - \sum_{s=1}^{E}g_{t,s}^i\right)\right\| + \frac{\sum_{j=1}^{d}L_j\left(x_t(j) - \bar{x}_t(j)\right)^2}{2} \tag{44b}$$

$$\leq \frac{\gamma 2^{2z}E^{2z+1}\sqrt{Q_z + G^{4z+2}}G}{2(2z+1)\sigma^{2z}} + \frac{\gamma^2 2^{4z}E^{4z+2}(Q_z + G^{4z+2})L_{\max}}{4(2z+1)^2\sigma^{4z}} + \frac{2\eta_z^2\gamma^2\sigma^2\sum_{j=1}^{d}L_j}{n}. \tag{44c}$$

$\square$

*Proof of Lemma 5.*

$$f(\bar{x}_t) - f(x_{t-1}) = f(\bar{x}_{t-1,E}) - f(\bar{x}_{t-1,0}) = \sum_{s=1}^{E}f(\bar{x}_{t-1,s}) - f(\bar{x}_{t-1,s-1}) \tag{45a}$$

$$\leq \sum_{s=1}^{E}\left(-\langle\nabla f(\bar{x}_{t-1,s-1}), \bar{x}_{t-1,s-1} - \bar{x}_{t-1,s}\rangle + \frac{L_{\max}}{2}\|\bar{x}_{t-1,s} - \bar{x}_{t-1,s-1}\|^2\right) \tag{45b}$$

$$= \sum_{s=1}^{E}\left(-\gamma\langle\nabla f(\bar{x}_{t-1,s-1}), \frac{1}{n}\sum_{i=1}^{n}g_{t-1,s}^i\rangle + \frac{\gamma^2 L_{\max}}{2}\|\frac{1}{n}\sum_{i=1}^{n}g_{t-1,s}^i\|^2\right). \tag{45c}$$

Taking expectation over gradient noise, we have

$$\mathbb{E}[\|\frac{1}{n}\sum_{i=1}^{n} g_{t-1,s}^i\|^2] \le \|\frac{1}{n}\sum_{i=1}^{n}\nabla f_i(x_{t-1,s-1}^i)\|^2 + \frac{\zeta^2}{n}, \tag{46a}$$

$$\mathbb{E}[-\langle\nabla f(\bar{x}_{t-1,s-1}), \frac{1}{n}\sum_{i=1}^{n} g_{t-1,s}^i\rangle] = -\langle\nabla f(\bar{x}_{t-1,s-1}), \frac{1}{n}\sum_{i=1}^{n}\nabla f_i(x_{t-1,s-1}^i)\rangle \tag{46b}$$

$$= -\frac{1}{2}\|\nabla f(\bar{x}_{t-1,s-1})\|^2 - \frac{1}{2}\|\frac{1}{n}\sum_{i=1}^{n}\nabla f_i(x_{t-1,s-1}^i)\|^2 \tag{46c}$$

$$+ \frac{1}{2}\|\nabla f(\bar{x}_{t-1,s-1}) - \frac{1}{n}\sum_{i=1}^{n}\nabla f_i(x_{t-1,s-1}^i)\|^2. \tag{46d}$$

Notice that from smoothness, we have for arbitrary $x, y \in \mathbb{R}^d$,

$$f(y) \le \langle\nabla f(x), y - x\rangle + \frac{L_{\max}}{2}\|y - x\|^2, \tag{47}$$

which is equivalent to

$$\|\nabla f(x) - \nabla f(y)\| \le L_{\max}\|y - x\|. \tag{48}$$

Now for every $s$, we have

$$\|\nabla f(\bar{x}_{t-1,s-1}) - \frac{1}{n}\sum_{i=1}^{n}\nabla f_i(x_{t-1,s-1}^i)\|^2 \tag{49a}$$

$$= \|\frac{1}{n}\sum_{i=1}^{n}\nabla f_i(\bar{x}_{t-1,s-1}) - \frac{1}{n}\sum_{i=1}^{n}\nabla f_i(x_{t-1,s-1}^i)\|^2 \tag{49b}$$

$$\le \frac{L^2}{n}\sum_{i=1}^{n}\|\bar{x}_{t-1,s-1} - x_{t-1,s-1}^i\|^2 \tag{49c}$$

$$= \frac{\gamma^2 L_{\max}^2}{n}\sum_{i=1}^{n}\left\|\sum_{q=1}^{s-1}\left(\frac{1}{n}\sum_{j=1}^{n} g_{t-1,q}^j - g_{t-1,q}^i\right)\right\|^2 \tag{49d}$$

$$\le \frac{(s-1)\gamma^2 L_{\max}^2}{n}\sum_{i=1}^{n}\sum_{q=1}^{s-1}\left\|\frac{1}{n}\sum_{j=1}^{n} g_{t-1,q}^j - g_{t-1,q}^i\right\|^2 \tag{49e}$$

$$\le 2(s-1)^2\gamma^2 L_{\max}^2 G^2. \tag{49f}$$

In all, we have

$$\mathbb{E}[f(\bar{x}_t) - f(x_{t-1})] \le \sum_{s=1}^{E}\left(-\frac{\gamma}{2}\|\nabla f(\bar{x}_{t-1,s-1})\|^2 - \frac{\gamma}{2}\|\frac{1}{n}\sum_{i=1}^{n}\nabla f_i(x_{t-1,s-1}^i)\|^2 + \frac{\gamma^2\zeta^2 L_{\max}}{2n}\right. \tag{50a}$$

$$\left. + \frac{\gamma}{2}\|\nabla f(\bar{x}_{t-1,s-1}) - \frac{1}{n}\sum_{i=1}^{n}\nabla f_i(x_{t-1,s-1}^i)\|^2 + \frac{\gamma^2 L_{\max}}{2}\|\frac{1}{n}\sum_{i=1}^{n}\nabla f_i(x_{t-1,s-1}^i)\|^2\right) \tag{50b}$$

$$\le \sum_{s=1}^{E}\left(-\frac{\gamma}{2}\|\nabla f(\bar{x}_{t-1,s-1})\|^2 + \frac{\gamma^2\zeta^2 L_{\max}}{2n} + (s-1)^2\gamma^3 L_{\max}^2 G^2\right) \tag{50c}$$

$$= -\frac{\gamma}{2}\sum_{s=1}^{E}\|\nabla f(\bar{x}_{t-1,s-1})\|^2 + \frac{E\gamma^2\zeta^2 L_{\max}}{2n} + \frac{E(E-1)(2E-1)\gamma^3 L_{\max}^2 G^2}{6}. \tag{50d}$$

$\square$

541 *Proof of Theorem 3.* We need a similar lemma like Lemma 4.

542 **Lemma 6.** *If $\sigma > E(G + Q_\infty)$, then*

$$\mathbb{E}[f(x_t) - f(\bar{x}_t)] \leq \frac{2\gamma^2\sigma^2 \sum_{j=1}^d L_j}{n}. \tag{51}$$

543 Following similar idea in the proof of Theorem 1, we have

$$\mathbb{E}[f(x_t) - f(x_{t-1})] = \mathbb{E}[f(x_t) - f(\bar{x}_t)] + E[f(\bar{x}_t) - f(x_{t-1})] \tag{52a}$$

$$\leq -\frac{\gamma}{2} \sum_{s=1}^E \|\nabla f(\bar{x}_{t-1,s-1})\|^2 + \frac{E\gamma^2\zeta^2 L_{\max}}{2n} + \frac{E(E-1)(2E-1)\gamma^3 L_{\max}^2 G^2}{6} + \frac{2\gamma^2\sigma^2 \sum_{j=1}^d L_j}{n}. \tag{52b}$$

544 Rearranging the terms, we have

$$\frac{1}{E} \sum_{s=1}^E \|\nabla f(\bar{x}_{t-1,s-1})\|^2 \leq \frac{2\mathbb{E}[f(x_{t-1}) - f(x_t)]}{E\gamma} + \frac{\gamma\zeta^2 L_{\max}}{n} + \frac{(E-1)(2E-1)\gamma^2 L_{\max}^2 G^2}{3} \tag{53a}$$

$$+ \frac{4\gamma\sigma^2 \sum_{j=1}^d L_j}{En}. \tag{53b}$$

545 Form the telescopic sum

$$\mathbb{E}[\frac{1}{TE} \sum_{t=1}^T \sum_{s=1}^E]\|\nabla f(\bar{x}_{t-1,s-1})\|^2 \leq \frac{2\mathbb{E}[f(x_0) - f^*]}{TE\gamma} + \frac{\gamma\zeta^2 L_{\max}}{n} + \frac{(E-1)(2E-1)\gamma^2 L_{\max}^2 G^2}{3} \tag{54a}$$

$$+ \frac{4\gamma\sigma^2 \sum_{j=1}^d L_j}{En}. \tag{54b}$$

546 Here we provide a simple example where $\sigma < E(G + Q_\infty)$ and the algorithm cannot converge.

Consider $E = 1$, $Q_\infty = 0$ and the problem

$$\min_{x \in \mathbb{R}} (x - A)^2 + (x + A)^2,$$

547 where $A > 0$ is some postive number. If we choose the initial to be $x_0 = \frac{A}{2}$. As we can, the gradient
548 at $x_0$ for the two parts of the objective function are $-A$ and $3A$ respectively. We denote that $\xi_\infty$ as
549 the random noise following uniform distribution at $[-1, 1]$. If now $\sigma < A$, we have

$$\text{Sign}(-A + \sigma\xi_\infty) + \text{Sign}(3A + \sigma\xi_\infty) = 0, \tag{55}$$

550 i.e., this algorithm never update the variable. $\qquad\square$

551 *Proof of Lemma 6.* We first note that, when $z = +\infty$, we have

$$\Psi_\infty(x) = \begin{cases} x & x \in [-1, 1], \\ 1 & x < -1, \\ 1 & x > 1. \end{cases} \tag{56}$$

552 Now, from $L$-smoothness we have,

$$f(x_t) - f(\bar{x}_t) \leq \langle \nabla f(\bar{x}_t), x_t - \bar{x}_t \rangle + \frac{\sum_{j=1}^d L_j (x_t(j) - \bar{x}_t(j))^2}{2}. \tag{57a}$$

553 The following equation and inequality can be checked, where the expectation is taken over $\xi_\infty$,

$$\mathbb{E}[x_t - \bar{x}_t] = \mathbb{E}\left[\frac{\gamma}{n} \sum_{i=1}^n \left(\sigma\text{Sign}\left(\sum_{s=1}^E g_{t,s}^i + \sigma\xi_\infty\right) - \sum_{s=1}^E g_{t,s}^i\right)\right] = 0, \tag{58}$$

554 because from the condition of $\sigma$ we can see that $\sigma > \|\sum_{s=1}^E g_{t,s}^i\|_\infty$ almost surely. $\qquad\square$

For any $j = 1, ..., d$, we have

$$\mathbb{E}[(x_t(j) - \bar{x}_t(j))^2] \leq \frac{\gamma^2}{n^2} \mathbb{E}\left[\left(\sum_{i=1}^n \left(\sigma \text{Sign}\left(\sum_{s=1}^E g_{t,s}^i(j) + \sigma\xi_\infty(j)\right) - \sigma\Psi_\infty\left(\frac{1}{\sigma}\sum_{s=1}^E g_{t,s}^i(j)\right)\right)\right)^2\right] \tag{59}$$

$$\leq \frac{4\gamma^2\sigma^2}{n}. \tag{60}$$

Hence, we have

$$\mathbb{E}[f(x_t) - f(\bar{x}_t)] \leq +\frac{\sum_{j=1}^d L_j \left(x_t(j) - \bar{x}_t(j)\right)^2}{2} \tag{61}$$

$$\leq \frac{2\gamma^2\sigma^2 \sum_{j=1}^d L_j}{n}. \tag{62}$$

## D  A simple simulated experiment.

In this section, we verify our theorical results in Section 3 on a simple simulated experiment without any gradient noise. Specifically, we consider the following distributed optimization problem with 10 clients,

$$\min_{x \in \mathbb{R}^d} \frac{1}{2} \sum_{i=1}^{10} \|x - y_i\|^2. \tag{63}$$

Here we generate $y_1, ..., y_{10} \in \mathbb{R}^d$ using i.i.d standard Gaussian distribution, where $d$ is the problem dimension. We compare the performance of the following algorithms. For all the algorithms, we use the same stepsize 0.01 and all-zero initialization. We denote the tested algorithms as:

- GD: Distributed gradient descent without any compression.
- Sto-SignSGD: The algorithm proposed by [Safaryan and Richtárik, 2021].
- SignSGD: (Algorithm 1 with $z = 1$, $E = 1$ and $\sigma = 0$.).
- 1-SignSGD (Algorithm 1 with $z = 1$ and $E = 1$.)
- $\infty$-SignSGD (Algorithm 1 with $z = +\infty$ and $E = 1$.)

**Results.** As we can see from Figure 3, all the stochastic sign-based algorithms can converge to the optimal solution, while the SignSGD without any noise fail to converge to the optimal solution. Besides, 1-SignSGD and $\infty$-SignSGD have roughly the same convergence speed which is slightly slower than the uncompressed gradient descent. It is also verified that the input-dependent noise scale adopted by [Safaryan and Richtárik, 2021] could lead to slow convergence when the problem dimension is high, as we have discussed in Section 3.2. The optimal noise scales of 1-SignSGD and $\infty$-SignSGD are selected based on Figure 4. We can see that there is a clear bias-variance trade-off in 4 which corroborates our prediction in Section 3. Moreover, it worth to mention that in this experiment, the optimal $\sigma$ for $\infty$-SignSGD is much smaller than the conservative choice suggested by theory.

## E  Experiment details

### E.1  Details for the experiment in Section 4.1

In Table 2, we provide the tuned hyperparameters for all the tested algorithms on non-i.i.d MNIST. Generally, we tune the hyperparameters via grid search: $[0.1, 0.05, 0.01, 0.005]$ for stepsize, $[0, 0.3, 0.5, 0.7, 0.9]$ for momentum coefficient, $[0, 0.02, 0.05, 0.01, 0.03, 0.05, 0.1, 0.3, 0.5]$ for noise scale.

In Figure 5, we visualize the performance of 1-SignSGD and $\infty$-SignSGD under different noise scales. As we can see, the results for 1-SignSGD and $\infty$-SignSGD are almost the same, except that the $\infty$-SignSGD is slighly better than 1-SignSGD when the noise scale is large.

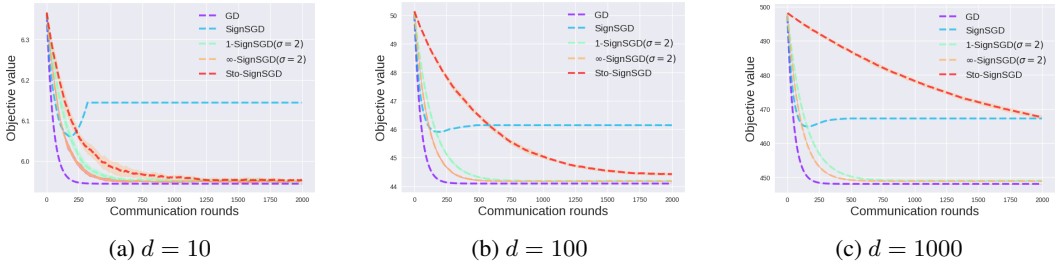

(a) $d = 10$    (b) $d = 100$    (c) $d = 1000$

Figure 3: Performance of algorithms under different problem dimension.

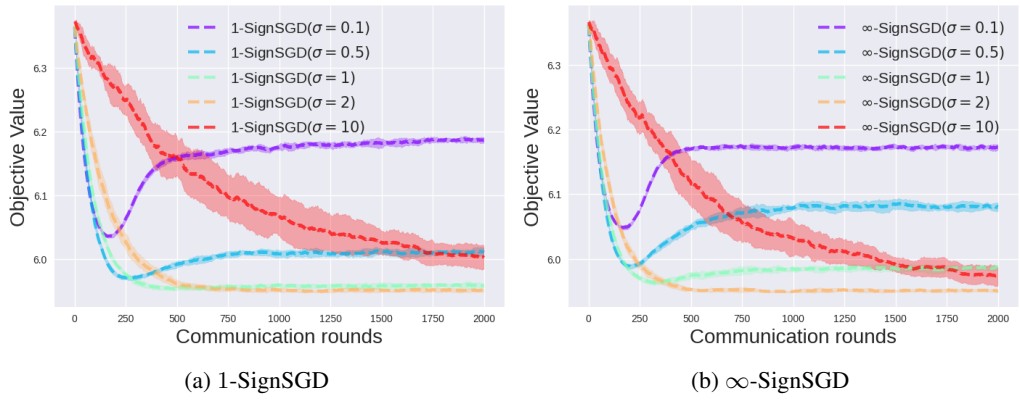

(a) 1-SignSGD    (b) $\infty$-SignSGD

Figure 4: Algorithm 1 under different noise scales

## E.2  Details for the experiment in Section 4.2

For the experiment on EMNIST, we fix the client stepsize as 0.05. Then we tune the server stepsize, noise scales via grid search: $[1, 0.5, 0.1, 0.05, 0.01, 0.005]$ for stepsize, $[0, 0.005, 0.02, 0.05, 0.01, 0.03, 0.05, 0.1, 0.2]$ for noise scale. The used hyperparameter in the Figure 2 are summarized in Table 3. We also visualize the performance of 1-SignFedAvg and $\infty$-SignFedAvg under various noise scales and local steps in Figure 6, 7, where we use SignFedAvg to represent Algorithm 1 with $\sigma = 0$.

| Algorithm | Stepsize | Momentum coefficient | Noise scale |
|:---:|:---:|:---:|:---:|
| SGDwM | 0.05 | 0.9 | |
| EF-SignSGDwM | 0.05 | 0.9 | |
| Sto-SignSGDwM | 0.01 | 0.9 | |
| SignSGD | 0.01 | 0 | 0 |
| 1-SignSGD | 0.01 | 0 | 0.05 |
| $\infty$-SignSGD | 0.01 | 0 | 0.05 |

Table 2: Hyperparameters for tested Algorithms on non-i.i.d MNIST.

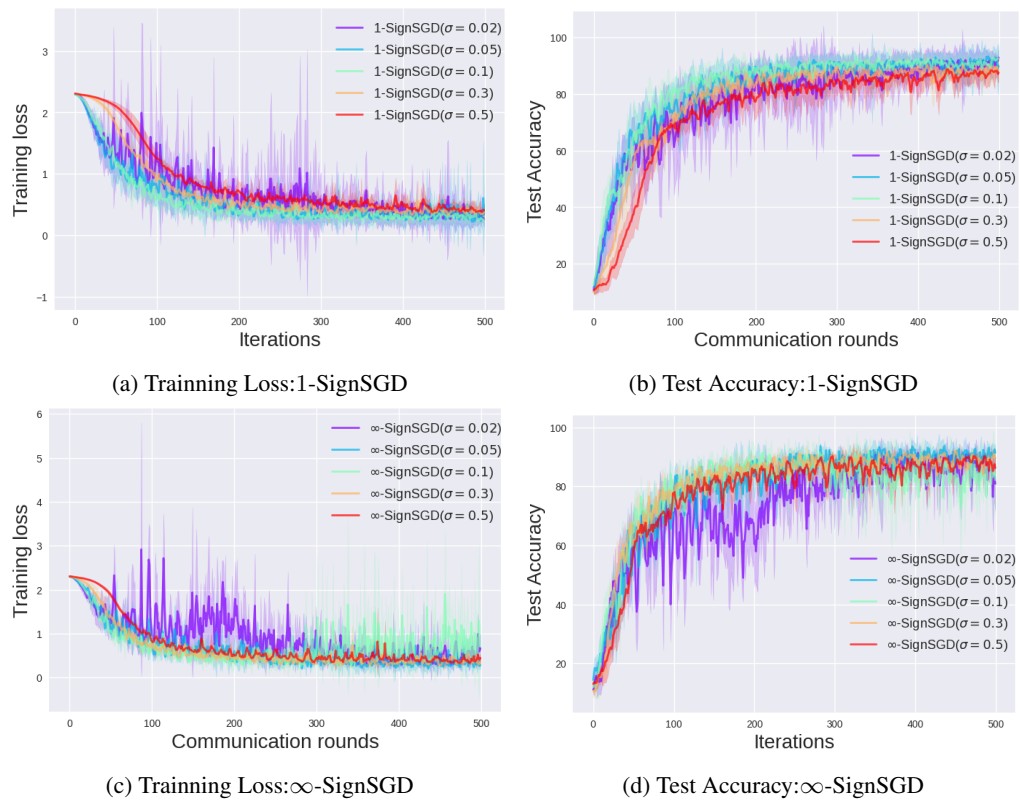

(a) Trainning Loss:1-SignSGD

(b) Test Accuracy:1-SignSGD

(c) Trainning Loss:∞-SignSGD

(d) Test Accuracy:∞-SignSGD

Figure 5: ALG 1 under different noise scales on non-i.i.d MNIST

| Algorithm | Server stepsize | Noise scale |
|---|---|---|
| 1-SignFedAvg | 0.03 | 0.01 |
| ∞-SignFedAvg | 0.03 | 0.01 |

Table 3: Hyperparameters for tested Algorithms on EMNIST.

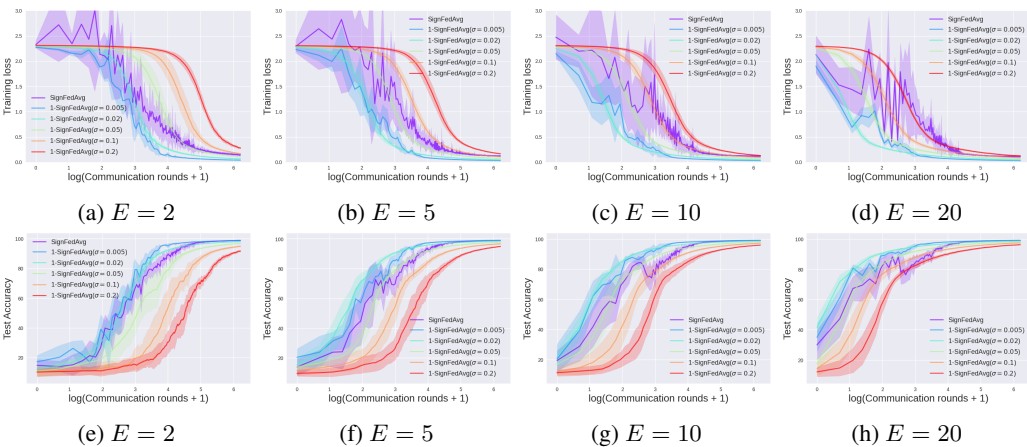

(a) $E = 2$

(b) $E = 5$

(c) $E = 10$

(d) $E = 20$

(e) $E = 2$

(f) $E = 5$

(g) $E = 10$

(h) $E = 20$

Figure 6: 1-SignFedAvg under different noise scales and local steps

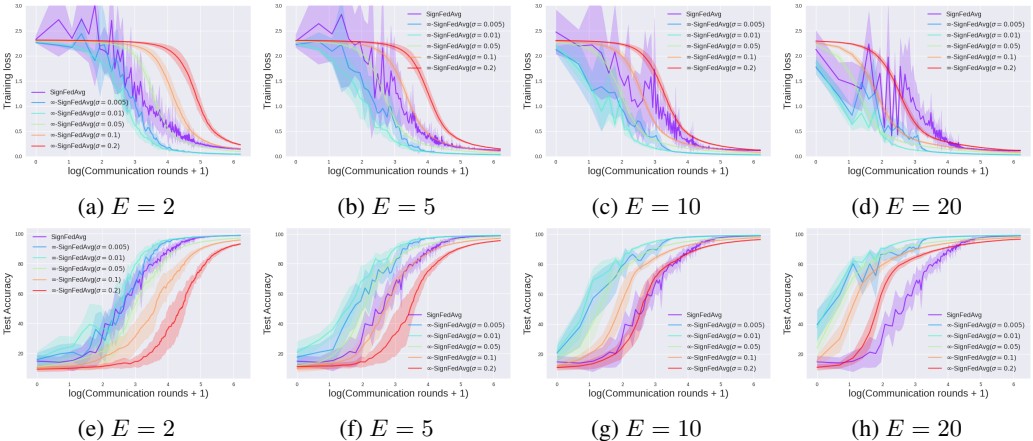

Figure 7: ∞-SignFedAvg under different noise scales and local steps

