# OpenReview forum: "$z$-SignFedAvg: A unified sign-based stochastic compression for federated learning"
_NeurIPS.cc/2022/Workshop/Federated_Learning — FL-NeurIPS 2022 Poster_

### Official Review · Reviewer_FaZd · 2022-10-12
**Interesting Paper**

The paper makes two contributions: (i) it presents a framework to analyze sign-based compression schemes for federated learning, where a noise perturbation is applied before the sign operator; (ii) it proposes SignFedAvg algorithm, which can be viewed as an extension of SignSGD in [Bernstein et al. 2018] to FedAvg.

The idea of SignFedAvg is quite natural in the context of SignSGD and its stochastic variants. While SignSGD applies sign-based compression to FedSGD, SignFedAvg applies sign-based compression to FedAvg (i.e., model updates). It also adds a noise perturbation to make the compression unbiased.

Strengths:
1. SignFedAvg algorithm is a natural extension of SignSGD, and easy to implement.
2. The paper is overall well-written.

Weaknesses:
1. My main concern is that, for Theorem 3 in appendix, the variance term is really large. The authors compare the convergence result with [Yu et al. 2019], but the variance term in that paper is proportional to the variance of SGD operator. On the other hand, in Theorem 3, the variance term equals $(4\sigma^2 \sum_{j=1}^{d}L_j)/(E (n\tau)^{1/2})$, where $\tau = TE$. Since $\sigma > E(G + Q_{\infty})$, the variance term turns out to be $(4 E G^2 \sum_{j=1}^{d}L_j)/ ((n\tau)^{1/2})$, which seems significantly large. Further, it is unclear to me why the variance term behaves as $O((n\tau)^{-1/2})$ when $E\leq n^{-3/4}\tau^{1/4}$ as claimed in Theorem 3. It would be important to add more details.

2. The paper says that their theoretical convergence guarantees are better than existing sign-based methods. The paper also claims that SignFedAvg is the first sign-based federated averaging algorithm. These two claims sound a bit contradictory, and further discussion will be helpful.

Additional comment:

1. In Fig.1, the training loss for SignSGD(\sigma = 0) keeps fluctuating, suggesting that the algorithm is probably not converging. However, the test accuracy seems to improve with communication rounds. It would be good to discuss why this might be happening.

---

### Official Review · Reviewer_JvG3 · 2022-10-17
**Review summary**

This paper introduces z-SignFedAvg, a gradient compression approach with z-distribution.

Strength:

(1) The proposal of z-distribution is novel, it provides a nice theoretical bound in the compression error, as shown in Lemma 1

(2) The z-SignFedAvg has theoretical guarantees on the bias and variance

Although I am impressed by the theoretical results, I would like to raise some questions about algorithms and evaluation.

(1) What is the computation complexity of z-Sign, how does it compare with SignSGD?

(2) How does z-SignFedAvg perform in the end-to-end training efficiency in federated optimization?

(3) Does the advantages of z-SignFedAvg scale to other datasets such as Shakespeare?

---

### Official Review · Reviewer_dxB3 · 2022-10-18
**Paper 17 review**

This work suggest a generalized version of stochastic sign-based compressor for distributed optimization. This method adds a special noise from so-called $z$-distribution (which covers Gaussian and uniform as special cases) and applies a sign operator on top of it resulting in an asymptotically unbiased compressor (up to a constant factor). Then the proposed algorithm is combined with local-SGD type method motivated by federated learning applications. The theoretical benefits of the latter are elusive for me as it was previously shown that in the heterogeneous case local methods have no benefit over simple baselines like mini-batch SGD. While it may be useful from practical perspective but the experimental evaluation is too limited to draw consistent conclusions on the benefits of the proposed method in realistic scenarios. Extending the simulations to at least one dataset like (label) heterogeneous CIFAR-10 would be more representable. In addition, I do not quite understand why other (not sign) quantization-based baselines are missing.

I would like to ask the authors to add a more detailed discussion on assumption 2 as it seems to be more restrictive than typically used conditions for variance of the stochastic gradients. Apart from it, Assumption 1 (A.4) is very strong and may not hold even for subsampled gradients for sum of quadratic functions over unbounded domain.

I advice the authors to use the spell-checker as there are a lot of simple typos throughout the text.

Besides, I noticed that a lot of works listed in references were not referred to in the main text and Appendix.

To conclude, I think that some of the paper’s ideas (proposed compressor) can be interesting for the community. However, it be improved from theoretical side by relaxing some of the assumptions (maybe for a different method without local steps) and/or in empirical part.

---

### Decision · Program_Chairs · 2022-10-20

Accept (Poster)